# Reducing Employees’ Time Theft through Leader’s Developmental Feedback: The Serial Multiple Mediating Effects of Perceived Insider Status and Work Passion

**DOI:** 10.3390/bs14040269

**Published:** 2024-03-24

**Authors:** Zhen Wang, Qing Wang, Daojuan Wang

**Affiliations:** 1School of Labor and Human Resources, Renmin University of China, Beijing 100872, China; wangzhen611@ruc.edu.cn; 2Aalborg University Business School, Aalborg University, 9220 Aalborg, Denmark; daw@business.aau.dk

**Keywords:** supervisor developmental feedback, time theft, perceived insider status, work passion, perceptions of leader hypocrisy, multiple mediating effect, topic analysis

## Abstract

Time theft, especially with the shift to remote work during the pandemic, is an increasing challenge for organizations. Existing studies demonstrate that both authoritarian leadership and laissez-faire leadership can exacerbate time theft, putting leaders in a behavioral dilemma of neither being strict nor lenient. Additionally, the pervasive and covert nature of time theft diminishes the effectiveness of subsequent corrective actions. Our study aims to investigate how to prevent time theft by mitigating employees’ inclinations. Based on role theory, our study examines whether supervisor developmental feedback can encourage employees to perform work roles more appropriately. To uncover the complicated internalization process of role expectation, our study incorporates perceived insider status and work passion as serial mediators and considers the boundary effect of leaders’ word–deed consistency. In Study 1, a survey of 402 employees revealed that supervisor developmental feedback can negatively predict employee time theft through employees’ perceived insider status and work passion. Study 2 employs the same sample to further identify three topics of supervisor developmental feedback: skill learning, attitude learning, and social learning. Moreover, serial multiple mediating effects are affirmed across topics. The findings suggest that providing feedback on employees’ learning and growth is an effective approach to prevent time theft.

## 1. Introduction

In 2023, a Canadian accountant was ordered to repay their former employer CAD 2600 for time theft tracked while working remotely [1]. With the COVID-19 pandemic accelerating the shift towards remote work, addressing employees’ time theft has emerged as a focal issue worldwide across industries and in behavioral science research. However, beyond monitoring and punishment, is it possible for leaders to encourage employees to actively reduce time theft?

Time theft refers to employees’ tendency to partake in unauthorized non-work-related activities during paid work hours, including behaviors such as arriving late, leaving early, making personal phone calls, or shopping online during work [2,3,4]. The term ’theft’ is used because employees are compensated for their work hours [5]. Research on time theft originates from two trajectories. First, as studies on counterproductive behaviors and workplace deviant behaviors evolve, scholars have begun to explore less harmful theft behaviors, including time theft [6,7]. Although a minority of studies suggest that a short unobtrusive break can help restore employees’ concentration [8], organizations typically view time theft as a form of productivity disruption or deviance [9,10,11]. Second, the early 21st-century saw rapid internet growth, enhancing workplace productivity but also sparking concerns over its negative impacts. Scholars from the psychology, management, and computer science fields delved into cyberloafing (also known as cyberslacking or cyberbludging), which involves employees engaging in personal internet-based activities during work hours [11,12,13,14]. Building on both trajectories, time theft emerges as an independent concept focused on comprehensive non-work-related behaviors rather than web-based off-task behaviors during work hours, potentially including cyberloafing, shirking, job neglect, social loafing, free riding, etc. [4,5]. The rise of remote work due to the COVID-19 pandemic has again prompted discussions on how to better manage employees’ time theft.

In general, there are two approaches to reducing time theft: preventive measures beforehand and corrective actions afterward. While punitive measures are often employed by organizations to deter employees from time theft, the effectiveness of this ’cat and mouse game’ is usually seen as unsatisfactory [5]. This is partly because time theft, unlike other counterproductive behaviors, is hard to detect [3], especially as the majority of employees disapprove of time theft tracking [15]. Additionally, strict control from leadership can further lead to employee emotional exhaustion, which in turn exacerbates time theft [16]. Moreover, in certain instances, addressing time theft can be more time-consuming than ignoring it due to the risk of legal complications [5]. Another method to reduce time theft is through preventive measures. Existing research has identified massive antecedents of time theft, which can be categorized into two types: one views time theft as a nonaggressive response to work-related antecedents, such as after-hours electronic communication [17], authoritarian leadership [16], and workplace ostracism [14], while the other sees time theft as a purposeful action, e.g., spending work time on personal matters [18]. Ideally, understanding and intervening in these antecedents can effectively mitigate time theft. However, the complexity of the antecedents makes intervention challenging. For example, both authoritarian and laissez-faire leadership can lead to employee time theft [16,19], placing leaders in a dilemma of not being too strict or too lenient. Moreover, the pervasive nature of time theft and the purposeful features of time theft in certain instances illustrate the difficulty of reducing it solely through cause control [2,5].

To address the limitations in current research on preventing employee time theft, our study leverages role theory to understand time theft from the perspective of role and explores how to encourage employees to perform their work roles in accordance with organizations’ expectations in the workplace. Role theory suggests that individuals juggle multiple roles [20], and employees’ time theft can be seen as a conflict between work and non-work roles. Existing studies have argued that the internalization of work role expectations by an employee is influenced not only by the centrality of their work role but also by role ambiguity [21]. Furthermore, information related to work roles is typically obtained through interactions with others [20], especially leaders, who are key figures in the workplace with close interactions with employees. In organizations, leaders often have a significant impact on shaping employee behavior through their feedback [22]. Leaders’ feedback can be classified into close monitoring feedback and developmental feedback, with the latter providing helpful and useful information to employees and assisting in their future learning and growth [23]. Previous studies indicate that developmental feedback can convey general information to employees beyond specific task information, including relational roles, cultural norms, and both in-role and extra-role expectations [24], and can additionally enhance employees’ intrinsic motivation by fostering personalized development [25,26]. Consequently, leaders’ developmental feedback has the potential to reduce role ambiguity and elevate the centrality of work roles among employees, ultimately mitigating their theft of work time.

To validate whether supervisor developmental feedback can prevent employee time theft, our study proposed a serial multiple mediation model to capture employees’ attitudinal reactions. Specifically, helpful information provided by leaders may enhance employees’ perception of being organizational insiders, subsequently improving their work passion and ultimately reducing time theft. Furthermore, as employees might question leaders’ word–deed inconsistency [27], our study introduced perceptions of leader hypocrisy as a boundary condition. The moderated serial multiple mediation model was tested in Study 1. Previous scholars have argued that research on the content of supervisor developmental feedback is lacking [28]. According to role theory [20,21], supervisor developmental feedback on different topics for the same behavior intensity may convey different role expectations to employees, thereby impacting employees’ responses. Hence, Study 2 aimed to further explore the topic structure of supervisor developmental feedback and validate the hypothesized model across various developmental feedback topics.

## 2. Theoretical Framework and Hypotheses

### 2.1. Role Theory Perspective: Supervisor Developmental Feedback and Employees’ Time Theft

Role, as a fundamental concept of role theory, refers to behavioral expectations placed on individuals based on their position in a social structure [20,21]. From the role theory perspective, individuals often have multiple roles. Although work roles are the dominant roles for individuals in the workplace, other individuals’ roles might be performed during work time, e.g., a family role to engage in leisure activities with families. However, as those non-work roles are generally incompatible or inconsistent with work roles in terms of their role expectations, time theft can be regarded as a status of role conflict. Previous studies have revealed that the rationale of role conflict lies in the centrality of different roles; when the centrality of non-work roles is stronger, employees may disengage more from work roles [20,21]. Thus, the behavioral choices employees make to conduct time theft essentially point to a trade-off between the centrality of work and non-work roles.

Feedback is not only a means for leaders to convey evaluative or corrective information to employees, but is also commonly used for motivating employees or intervening in employees’ behavior within organizations [29]. In contrast to close monitoring of employees’ performance, supervisor developmental feedback is a form of feedback through which leaders provide helpful or valuable information to employees that enables them to learn, develop, and make progress on the job [23]. According to role theory, the concept of roles and their centrality can be influenced by social interaction [21]. This study proposes that supervisor developmental feedback may reduce work role ambiguity and increase employees’ work role centrality, further motivating employees to conduct less time theft. First, future-oriented informational feedback is vital for communicating organizational role prototypes to employees. Work role ambiguity describes employees who are not clear on role boundaries or how to get ahead [21]. According to role theory, employees tend to look at current work for role identification as well as at their overarching work role [30]. While performance feedback emphasizes the completion of specific performance goals [23], developmental feedback from leaders provides employees with more general information and focuses more on developing abilities (e.g., technical guidance and social norms), which enhances employees’ big-picture thinking of in-role and extra-role expectations over the relatively long term [24]. Developmental feedback supplements performance feedback on employees’ future guidance, thereby reducing employees’ work role ambiguity. For instance, certain employees may perform time theft due to feeling overqualified about work tasks [31]; supervisor developmental feedback can offer the possibility of releasing personal potential by providing learning opportunities. Second, supervisor developmental feedback may enhance the centrality of employees’ work roles. Role centrality refers to the “importance people give to roles central to their life and identity” [21]. Previous studies have affirmed that helpful and valuable information communicated by supervisor developmental feedback can boost an individual’s intrinsic motivation, which emphasizes employees’ own interest and pleasure in work [25,26]. With a higher level of intrinsic motivation, employees may value their work more and tend to increase their work role centrality. This greater level of centrality in work roles predicts increased input of effort for a given job [21], making it more likely that employees may proactively conduct less time theft. Thus, this study proposes the following hypothesis: 

**Hypothesis 1.** 
*Supervisor developmental feedback can reduce employees’ time theft.*


### 2.2. Perceived Insider Status as a Mediator between Supervisor Developmental Feedback and Time Theft

According to role theory [21], a prerequisite for analyzing employees’ work roles is that employees are aware that their work roles belong to them. However, employees who physically work in organizations may experience differential perceptions of insider status [32]. This pair of opposing concepts, perceived organizational insider and perceived organizational outsider, further divides work roles based on the psychological relationship between employees and organizations, and lays the foundation for understanding the rationale behind employees’ work attitudes and behavior differences.

Leaders are commonly regarded as representatives of organizations who have the right to distinguish organizational insiders and outsiders [32]. Prior scholars have argued that the perception of leader support can enhance employees’ sense of being organizational insiders [33,34]. Supervisor developmental feedback can provide useful information regarding employees’ future growth instead of focusing only on short-term performance [23], thereby communicating a signal of leaders’ support and attention [35]. Recognizing leaders’ support from developmental feedback, employees may develop their role as organizational insiders. In addition, supervisor developmental feedback creates a work environment of freedom, equality, and autonomy, which may enhance employees’ perceptions of being organizational insiders through trust-based inducements [36]. When they feel regarded as organizational insiders, employees may prioritize organizational interests and be willing to sacrifice personal interests [37]. Hence, employees may attempt to align their actions with organizational expectations. Specifically, they might improve task performance [37] and even perform more organizational citizenship behavior [36]. Therefore, such employees are likely to have stronger willingness to conduct more self-regulation and to perform less time theft in order to ensure organizational benefit. Thus, this study proposes the following hypothesis: 

**Hypothesis 2.** 
*Employees’ perceived insider status mediates the relationship between supervisor developmental feedback and employees’ time theft.*


### 2.3. Work Passion as a Mediator between Supervisor Developmental Feedback and Time Theft

Work passion is defined as a strong tendency towards work that individuals enjoy, find meaningful, and identify with, and to which they commit certain time and other resource [38]. Employees’ work passion is not changeless, and can be influenced by external factors [39]. In organizations, employees frequently interact with their leaders at work; thus, feedback from leaders has a significant impact on how employees behave [22]. While leaders’ feedback can be categorized into controlling feedback and developmental feedback, controlling feedback involves closely monitoring employees’ task performance, potentially undermining their intrinsic motivation; on the other hand, developmental feedback is a form of informational feedback that emphasizes long-term support and improvement, fostering positive emotions and intrinsic motivation towards work among employees [23,25,26]. Based on role theory, supervisor developmental feedback not only offers employees more knowledge about future growth, which reduces their long-term role ambiguity, but also delivers messages on the meaning of work. When employees acknowledge their leaders’ attention and support, they may increase their work role centrality and even internalize it as a part of their self-concept. Therefore, employees who experienced more supervisor developmental feedback may manifest higher levels of work passion.

Moreover, existing studies have proved that work passion such as a hardworking attitude has a crucial impact on employees’ action choices to achieve work goals [40]. Time theft is a relatively low-harm and not entirely intentional deviant behavior within an organization [5], which in many cases is caused by emotional exhaustion [41,42]. However, work passion holds the opposite status, which encompasses the experience of joy as an emotional component within the concept [39]. Employees with high work passion tend to invest more resources in work and gain higher job satisfaction [43]. Further, employees may actively reduce their time theft. Thus, this study hypothesizes that: 

**Hypothesis 3.** 
*Employees’ work passion mediates the relationship between supervisor developmental feedback and employees’ time theft.*


### 2.4. Perceived Insider Status and Work Passion as Serial Multiple Mediation between Supervisor Developmental Feedback and Time Theft

Role theory contends that the process of an individual behaving in a role is the process of internalizing the role expectations [20,21]. For the combination of supervisor developmental feedback and perceived insider status together to describe employees’ cognition towards roles and role expectations, it remains unanswered how the work role centrality changes in employees’ self-concept. For instance, existing studies have affirmed that both supervisor developmental feedback and perceived insider status have a positive relationship with employees’ organizational citizenship behavior [36,44]. However, employees’ engagement in organizational citizenship behavior might reduce employees’ time input in their own work, resulting in a form of time theft which is described as work-engaged but unproductive [2]. Thus, it is possible that employees who have received frequent developmental feedback from leaders and are perceived as organizational insiders not only increase inputting efforts in their work but strengthen other organizational or non-organizational roles as well. This leads to an uncertain change in the status of work roles in employees’ personal concept. Thus, our study further links perceived insider status and work passion in order to narrow the scope of work role expectations with respect to personal work status and outcomes (which corresponds to the ’work’ scope in the concept of time theft) and explore the reactive changes in employees’ work role centrality. An employee with work passion may experience a motivational love toward work and exhibit high levels of persistence and effort towards it, such that the work ultimately becomes a part of the employee’s identity [40]. Empirical research affirms that perception as an organizational insider significantly influences employees’ assessment of their work responsibilities [45]. Work passion is a dynamic emotional experience [46]. When employees feel that their work roles have been valued by organizations, they may in turn value the work as an integral part of their self-concept. In other words, employees’ perceived insider status might increase their work passion. Moreover, as mentioned above, employees may perform less time theft. Thus, this study hypothesizes that: 

**Hypothesis 4.** 
*Employees; perceived insider status and work passion play serial mediating roles between supervisor developmental feedback and employees’ time theft.*


### 2.5. Perceptions of Leader Hypocrisy as a Moderator

Employees tend to view leaders with inconsistent words and actions as hypocritical [47], which is linked to various negative attitudinal and behavioral outcomes, e.g., turnover intentions [27] and deviant work behaviors [48]. Considering the hypocritical features of leaders, previous studies have initiated preliminary explorations into the ambivalent impact of leadership styles. For example, scholars have argued that when employees attribute leadership humility to impression management tactics, i.e., leaders’ attempts to be seen favorably or positively, they will perceive leaders as hypocritical and may even perform more time theft [49]. Additionally, when leaders’ self-sacrifice is attributed to low authenticity, employees may view leaders as hypocritical and lose trust in them [50]. Thus, our study further explores the boundary effect of perceptions of leader hypocrisy. Supervisor developmental feedback provides employees with future-oriented information [23]. When employees detect incongruence between leaders’ words and deeds, doubts may arise about the sincerity of their developmental feedback, leading employees to question whether it is merely impression management tactics [49]. Previous research has proven that leader inconsistency may induce employees to behave inconsistently themselves [47], possibly resorting to subtle forms of retaliation, e.g., knowledge-hiding [51]. Hence, perceptions of leader hypocrisy may undermine the mitigation effects of supervisor developmental feedback on employees’ time theft. As a potential result of leader hypocrisy, employees might perceive that the organization does not truly consider them insiders, leading them to question the certainty of future support and assistance from leadership [27]. Ultimately, employees may be inclined to maintain the status quo in their work, and might not trigger a higher level of work passion. Additionally, when a high level of leadership hypocrisy is perceived, employees may think that leaders themselves cannot lead by example in terms of avoiding time theft [5]. Further, employees will feel less motivated to reduce time theft and tend to stay in their current situation. Therefore, this study proposes the following hypothesis: 

**Hypothesis 5.** 
*Perceptions of leader hypocrisy weaken the positive impact of supervisor developmental feedback on perceived insider status.*


**Hypothesis 6.** 
*Perceptions of leader hypocrisy weaken the influence of supervisor developmental feedback on reducing employees’ time theft through the serial mediating effects of perceived insider status and work passion.*


## 3. Study 1 Quantitative Study to Validate the Serial Multiple Mediation Model

### 3.1. Participants

Study 1 employed a quantitative research paradigm. A convenience sampling method was adopted to recruit currently employed individuals to participate in the survey. During the questionnaire collection stage, a total of 600 online questionnaires were distributed, of which 594 were returned. After excluding invalid questionnaires with missing data or inconsistent responses, the remaining 402 valid questionnaires were used for further analysis, resulting in a valid response rate of 67.0%. Before participants took the survey, this study provided standardized instructions, making sure to explain why the survey was being conducted and how it would proceed. Additionally, it was highlighted that any information shared by respondents would be confidential and only be used for academic purposes. To acknowledge participation, respondents were provided with an electronic red envelope containing approximately 0.5 US dollars.

The sample’s demographic features include gender, age, education, job type, industry, tenure, and job level. In summary: (a) 64.2% were female and 35.8% male; (b) regarding age, the majority were in the 20–30 age group (51.2%), followed by the 30–40 age group (37.3%); (c) in terms of education, most held a bachelor’s degree (71.6%), with master’s degree holders being the second largest group (13.7%); (d) concerning job type, a significant majority of employees worked in private enterprise (62.2%); (e) for work experience, the most common range was 4 to 6 years (28.6%); (f) in terms of job levels, there were more regular employees (46.3%) and junior managers (22.1%); and (g) in terms of industry, the sample distribution revealed a relatively balanced representation, featuring a significant presence in the Internet, Information Technology, and Gaming sectors (14.9%), Transportation, Logistics, Trade, and Retail sectors (10.7%), and Healthcare sector (10.4%).

### 3.2. Measurement Tools

To ensure the reliability and validity of the measurement tools, all of the scales used in this study were authoritative mature scales previously validated in the literature. For Chinese respondents, English scales were translated using the “translation–backtranslation” procedure proposed by Brislin to ensure the accuracy of the scales [52]. All scales used a seven-point Likert scale, with 1 to 7 respectively representing “strongly disagree” to “strongly agree”. A few scales incorporated reversed items to control response style bias. In summary, all scales utilized in this study have been proven to have satisfactory reliability, with Cronbach’s α coefficient consistently exceeding 0.7. For all scale items, see Appendix A.

#### 3.2.1. Supervisor Developmental Feedback

This study employed the Supervisor Developmental Feedback Scale developed by Zhou [23]. Respondents self-reported their perception of supervisor developmental feedback, which comprised three items, e.g., “While giving me feedback, my supervisor focuses on helping me to learn and improve”. The second item of the scale was subject to reverse scoring. The reliability of this scale was good, with a Cronbach’s α coefficient of 0.711.

#### 3.2.2. Time Theft

Based on Lorinkova and Perry’s methodology for measuring employees’ time theft [53], this study utilized a three-item scale adapted from Bennett and Robinson’s Workplace Deviance Scale [6]. For example, “I have taken an additional or longer break than is acceptable at workplace”. In this study, the time theft scale demonstrated good reliability, with a Cronbach’s α coefficient of 0.772.

#### 3.2.3. Perceived Insider Status

This study employed a six-item Perceived Insider Status Scale developed by Stamper and Masterson, e.g., “My work organization makes me believe that I am included in it” [32]. Three items (the third, fourth, and sixth items) were subject to reverse scoring. The reliability of this scale was satisfactory in this study, with a Cronbach’s α coefficient of 0.823.

#### 3.2.4. Work Passion

This study adopted the Work Passion Scale developed by Vallerand et al., consisting of nineteen items [38]. Among these, fourteen items were employed to measure employees’ current levels of harmonious work passion and obsessive work passion, while the remaining five served as criteria to assess whether an employee had work passion or not. Considering our research objective of introducing the work passion concept in order to explore how much employees value their work, rather than comparing two types of work passion, the latter five items of the Work Passion Scale were employed (for example, “I invest a lot of time and energy into my work”). The fifth item was scored reversely. The scale had good reliability in this study, with a Cronbach’s α coefficient of 0.743.

#### 3.2.5. Perceptions of Leader Hypocrisy

This study employed the Perceptions of Leader Hypocrisy Scale developed by Dineen, Lewicki, and Tomlinson, comprising four items, e.g., “I wish my supervisor would practice what he or she preaches more often” [48]. The Perceptions of Leader Hypocrisy Scale demonstrated good reliability in this study, and the Cronbach’s α coefficient of the scale was 0.720.

#### 3.2.6. Control Variables

Drawing on prior empirical research on time theft [4,16,19,53], this study considered seven variables—gender, age, education, job type, industry, tenure, and job level—as control variables to minimize confounding factors and reduce potential biases in the study results. Specifically, gender was controlled using a binary variable (0 = male, 1 = female); age was categorized into five levels, ranging from 19 years and below to 51 years and above; education was divided into five categories, ranging from high school and below to doctoral graduate; job type was categorized into five classes (state-owned enterprises, public institutions, civil servants, private enterprises, and foreign-owned companies); industry was classified into thirteen categories (e.g.,financial industry); tenure was divided into five stages, ranging from less than one year to ten years and above; and job level was categorized into four levels: regular employees, junior managers, mid-level managers, and senior managers.

### 3.3. Results

#### 3.3.1. Harman’s One-Factor Test

Common Method Bias (CMB) refers to the artificial covariance between predictor and criterion variables caused by common raters, item characteristics, item context (e.g., context-induced mood), and measurement context (e.g., predictor and criterion variables measured at the same point in time) [54,55]. In this study, to minimize the impact of CMB, methods such as anonymous self-assessment by employees and online surveys were employed, with the aim of separating measurements spatially and psychologically and protecting the respondents’ anonymity.

Because the data of this study were collected from a single source (i.e., employees’ self-reports), Harman’s one-factor test, which has been commonly used in behavioral integrity and workplace deviance studies, was further employed to address the issue of CMB [27]. If a single factor emerges that explains a significant portion of the covariance among the variables, this suggests the presence of a common method factor [54]. This study employed SPSS 27.0 to conduct Harman’s one-factor testing. The results from the unrotated exploratory factor analysis reveal that the covariance explained by the first factor is 38.87%, which is below the 40% threshold. This finding suggests that no single factor in this study accounts for a substantial proportion of the total variance, indicating limited influence of CMB on the results.

#### 3.3.2. Confirmatory Factor Analysis

To further examine the discriminant validity across supervisor developmental feedback, time theft, perceived insider status, work passion, and perceptions of leader hypocrisy, this study conducted confirmatory factor analysis (CFA) [56]. Utilizing Mplus 8, this study constructed various factor models based on concepts of variables. The CFA results reveal that the model fit of the five-factor model tends to be most satisfactory, outperforming alternative factor models (see Table 1). The results indicate that the discriminant validity of this study is good and suitable for hypothesis testing.

#### 3.3.3. Descriptive Statistics and Correlation Analysis

Table 2 displays the means, standard deviations, and inter-correlations among the measured variables in Study 1. The results indicate a significant negative correlation between supervisor developmental feedback and employee time theft (r=−0.51, p<0.01). In addition, there is a significant positive correlation between supervisor developmental feedback and employees’ perceived insider status (r=0.64, p<0.01). Employees’ perceived insider status is significantly positively correlated with employees’ work passion (r=0.70, p<0.01), and employees’ work passion is significantly negatively correlated with employees’ time theft (r=−0.67, p<0.01). These findings preliminarily support the relationships among the main variables in the proposed serial multiple mediation model, laying the groundwork for a more in-depth examination of the research hypotheses.

#### 3.3.4. Hypothesis Testing on the Serial Multiple Mediating Effect

After controlling for gender, age, education level, job type, industry, tenure, and job level, this study employed supervisor developmental feedback as the independent variable, perceived insider status and work passion as mediating variables, and time theft as the dependent variable, then conducted hierarchical regression analysis using SPSS 27.0. The results in Table 3 reveal that supervisor developmental feedback has a significant negative impact on time theft (Model 6, β = −0.487, p<0.001), providing support for Hypothesis 1. Additionally, supervisor developmental feedback positively influences perceived insider status (Model 1, β = 0.598, p<0.001) and perceived insider status negatively influences time theft (Model 7, β = −0.604, p<0.001). When taking both supervisor developmental feedback and perceived insider status into account in the regression model, the results reveal that perceived insider status has a significant negative impact on time theft (Model 8, β = −0.481, p<0.001) and supervisor developmental feedback continues to have a significant negative effect on time theft (Model 8, β = −0.199, p<0.001). Thus, perceived insider status plays a partial mediating role in the relationship between supervisor developmental feedback and time theft, supporting Hypothesis 2. The hypothesis testing for the mediating role of work passion between supervisor developmental feedback and time theft followed a similar process. Supervisor developmental feedback positively predicts work passion (Model 3, β = 0.533, p<0.001) and work passion negatively predicts time theft (Model 9, β = −0.668, p<0.001); moreover, when taking both supervisor developmental feedback and work passion into account in the regression model, the results show that work passion has a significant negative on time theft (Model 10, β = −0.564, p<0.001) and the impact of supervisor developmental feedback on time theft remains significantly negative (Model 10, β = −0.186, p<0.001), supporting Hypothesis 3. In addition, the results of Model 11 affirm the serial multiple mediating effect between perceived insider status and work passion, preliminarily supporting Hypothesis 4.

Then, this study further employed Model 6 from the SPSS PROCESS macro proposed by Hayes and conducted 5000 bootstrap resamples to validate the serial multiple mediating effects [57]. The results in Table 4 reveal that the mediating effect of employees’ perceived insider status on the relationship between supervisor developmental feedback and employee time theft is significant (Effect = −0.288, 95% CI = [−0.376, −0.210]). The mediating effect of employees’ work passion on the relationship between supervisor developmental feedback and employee time theft is significant as well (Effect = −0.301, 95% CI = [−0.375, −0.236]). Furthermore, employees’ perceived insider status and work passion serve as a serial multiple mediating effect between supervisor developmental feedback and employee time theft (Effect = −0.382, 95% CI = [−0.477, −0.302]); thus, Hypothesis 4 is confirmed.

#### 3.3.5. Hypothesis Testing on the Moderating Effect of Perceptions of Leader Hypocrisy

To test the moderating effect of perceptions of leader hypocrisy on the relationship between supervisor developmental feedback and perceived insider status, this study utilized SPSS PROCESS macro Model 1 and conducted 5000 bootstrap resamples [57]. As shown in Table 2, perceptions of leader hypocrisy negatively predict perceived insider status (Model 2, β = −0.185, p<0.001). However, the interaction term between supervisor developmental feedback and perceptions of leader hypocrisy on the negative impact on perceived insider status is not significant (Model 2, β = −0.042, 95% CI = [−0.103, 0.018]). Therefore, this study rejects Hypothesis 5. As Hypothesis 5 is a prerequisite for Hypothesis 6, this study preliminarily rejects Hypothesis 6. Furthermore, this study employed SPSS PROCESS macro Model 83 to evaluate the significance of the moderation effect on the serial multiple mediation model [57]. The results in Figure 1 reveal that the difference in the moderation effect of perceptions of leader hypocrisy on the serial multiple mediation relationship between supervisor developmental feedback, perceived insider status, work passion, and time theft is not significant (95% CI = [−0.010, 0.032]). Consequently, Hypothesis 6 of this study is rejected.

## 4. Study 2 Topic Analysis on Supervisor Developmental Feedback Text and Serial Multiple Mediating Effects Validation

### 4.1. Background

Study 1 affirms that supervisor developmental feedback can reduce employees’ time theft via the serial multiple mediating effects of perceived insider status and work passion. Study 1 employed the Supervisor Developmental Feedback Scale proposed by Zhou to measure the intensity of developmental feedback behavior [23]. As research on supervisor developmental feedback has progressed over the past twenty years, scholars have pointed out that existing studies neglect the content analysis of supervisor developmental feedback [28].

Supervisor developmental feedback has specific content topics. As a form of feedback, supervisor developmental feedback has an interventive or corrective impact on employees’ behavior [29]. Although scholars have distinguished the concept of supervisor developmental feedback from close monitoring, emphasizing its feature of being future-oriented instead of setting mandatory performance goals [23], the useful information provided by leaders in conversation is often based on future directions or action plans. Yet, supervisor developmental feedback highlights a feeling of autonomy instead of closing control [24], and the action plans conveyed by supervisor developmental feedback are more likely to be voluntary and long-term in contrast to mandatory performance goals.

Different content topics of supervisor developmental feedback may have varying effects on employees’ perceptions. Drawing from the role theory perspective, supervisor developmental feedback covering various topics, even at the same level of behavioral intensity, may convey different role expectations to employees, thereby influencing their responses.

In current research, the concept of supervisor developmental feedback primarily emphasizes its features of being helpful and useful along with the goal of cultivating employees’ long-term capabilities. However, research regarding content topics has been overlooked in existing studies. Therefore, the purpose of Study 2 was to explore the textual topic structure of supervisor developmental feedback and to further validate the serial multiple mediation model established in Study 1 under different topics.

### 4.2. Participants and Instruments

Study 2 and Study 1 shared the same research sample. Based on the definition of supervisor developmental feedback proposed by Zhou [23], participants were presented with an open-ended written question: “Please recall the last time your supervisor provided you with helpful or valuable information that enabled you to learn, develop, and make improvements on the job, then reproduce the content of the supervisor’s feedback (i.e., what your supervisor said at that time, at least 10 words)”. To ensure response quality, participants were required to provide responses of at least ten words. Additionally, considering respondents’ memory cycles [53], this study only asked participants to recall the content of the most recent supervisor developmental feedback.

### 4.3. Procedures

This study adopted the framework proposed by Ruan and Huang, which integrates Sentence-Bidirectional Encoder Representations from Transformers (SBERT) and Latent Dirichlet Allocation (LDA) for identifying textual themes in supervisor developmental feedback [58]. LDA is fundamentally a three-layer Bayesian probability model, consisting of a document layer, topic layer, and feature word layer [59]. The basic idea of LDA is that a document is formed by a random mixture of latent topics, with each document’s topics corresponding to specific feature word distributions [60]. While the LDA model has become a mainstream method for text modeling in the field of machine learning and is widely used for mining text topics, it faces challenges when dealing with short text samples due to a lack of necessary contextual information in word co-occurrence, resulting in incoherent text topics and non-ideal topic recognition. Sentence-Bidirectional Encoder Representations from Transformers (SBERT) addresses these limitations by combining sentence context information with thematic features [58].

The overall process of identifying topics in supervisor developmental feedback text using the integrated SBERT and LDA framework involved four stages (see Figure 2): first, text preprocessing, including cleaning text data, removing stop-words, and tokenization; second, text vectorization, which employed the LDA topic model to calculate probabilities for the supervisor developmental feedback corpus and used the SBERT model to compute sentence embedding vectors for the feedback text; third, vector concatenation, where the probability vectors generated by the LDA model and the sentence embedding vectors from SBERT were concatenated using an encoder in the latent space to reconstruct the data; and finally, clustering and topic word extraction employing the K-means clustering method and extracting topic words from the clusters of supervisor developmental feedback. This study used Python 3.10 for text topic analysis.

#### 4.3.1. Data Preprocessing

First, this study eliminated high-frequency meaningless words from the text using stop-words. Next, part-of-speech filtering was performed on the feedback text. Considering that supervisor developmental feedback contains information about employees’ future learning and growth in the organizational context [23], without specifying a particular technical domain, this study used the ‘noun’ identification setting during part-of-speech filtering. Finally, this study used the Jieba segmentation component for tokenization processing of supervisor developmental feedback text, forming the text corpus of supervisor developmental feedback.

#### 4.3.2. Vectorization of Supervisor Developmental Feedback Text Based on SBERT and LDA

First, this study loaded the language model ‘distilbert-base-nli-mean-tokens’ as the pretrained model for supervisor developmental feedback text, vectorizing 402 pieces of supervisor developmental feedback text data and resulting in a 402 × 768 matrix of vectorized sentence embeddings for supervisor developmental feedback text. Second, to obtain the topic embedding vectors for supervisor developmental feedback text, the study utilized the LDA topic model for computation. To determine the optimal number of topics [58], the study considered both topic perplexity and topic coherence as evaluation dimensions (see Figure 3) and explored the topic numbers from 1 to 10. The calculations revealed that when the number of topics was 3, the topic perplexity showed the first inflection point while the topic coherence simultaneously exhibited a peak. Therefore, for this study we selected three topics for the LDA topic model computation. Finally, this study performed autoencoder training and used the t-distributed Stochastic Neighbor Embedding dimension reduction technique, using the sentence vectors formed based on SBERT and the topic vectors generated by the LDA model as input, then mapping them into vectors in a low-dimensional space.

#### 4.3.3. Topic Identification for Supervisor Developmental Feedback Text

The vectors in the low-dimensional space need further clustering to establish semantic clusters for topic identification. Thus, this study employed the K-means algorithm for topic clustering. This involves partitioning data points into K clusters, where each cluster contains the data points closest to its centroid, thereby maximizing the similarity among data points within the cluster and minimizing the similarity between data points in different clusters. Previous research indicates that the distribution of combined vectors derived from the SBERT+LDA approach is influenced by the number of topics in LDA topic modeling [58]. Thus, for this study we set K to 3. For the results of the topic clusters distribution in three dimensions, see Figure 4.

### 4.4. Results

#### 4.4.1. Topics Analysis from Supervisor Developmental Feedback Text

Table 5 presents the three topics obtained from the topic analysis of the supervisor developmental feedback text based on the SBERT-LDA framework: skill learning-oriented supervisor developmental feedback, attitude learning-oriented supervisor developmental feedback, and social learning-oriented supervisor developmental feedback. In terms of skill learning, supervisor developmental feedback provides employees with learning advice regarding specific skills or abilities, such as “Learning the application of ChatGPT to improve work efficiency”, “Cultivating customer communication and negotiation skills”, and “Improving writing skills by studying others’ expressions”. Regarding attitude learning, leaders provide guidance on employees’ future growth through cultivating attitudes, such as “Take your time to gradually adapt to the new environment…”, “Maintain a mindset of continuous learning”, and “Young individuals should take steady steps, stay calm, and the prospects for future development will be more promising”. Finally, social learning-oriented supervisor developmental feedback refers to leaders encouraging employees to learn from others or enhance teamwork, such as “Try to learn from outstanding colleagues to continuously strive for improvement, and enhance your professional qualities”, “Learn to communicate with colleagues, build good relationships, and harness everyone’s strengths”, and “Be more proactive in cooperation within the team, help colleagues with lower performance, and collectively improve team performance”.

#### 4.4.2. Validation of Serial Multiple Mediation Model with Supervisor Developmental Subtopics

Based on the results of the topic analysis of supervisor developmental feedback, this study further validated the hypotheses regarding the moderated serial multiple mediation model using skill learning-oriented supervisor developmental feedback, attitude learning-oriented supervisor developmental feedback, and social learning-oriented supervisor developmental feedback. As the hypothesis testing process aligns with Study 1, we present only the key findings here. For the detailed results of our hierarchical regression analysis and mediating effects using the bootstrap method, see Appendix B.

The results in Table 6 demonstrate that the weakening effect on employees’ time theft through a serial multiple mediating process of employees’ perceived insider status and work passion is significant across skill learning-oriented supervisor developmental feedback (β = −0.379, 95% CI = [−0.536, −0.253]), attitude learning-oriented supervisor developmental feedback (β = −0.323, 95% CI = [−0.461, −0.194]), and social learning-oriented supervisor developmental feedback (β = −0.352, 95% CI = [−0.552, −0.215]). However, the direct effect of skill learning-oriented supervisor developmental feedback on time theft remains significant after introducing both mediators (β = −0.125, 95% CI = [−0.282, 0.033]), whereas attitude learning-oriented supervisor developmental feedback and social learning-oriented supervisor developmental feedback are insignificant. This may suggest that the serial multiple mediating mechanism has limitations in explaining the relationship between skill learning-oriented supervisor developmental feedback and employees’ time theft. Additionally, the data in our study do not support the moderation effect of perceptions of leader hypocrisy in the relationship between supervisor developmental feedback and perceived insider status across three topics of supervisor developmental feedback.

## 5. Discussion

### 5.1. Conclusions

The results from Study 1 prove that supervisor developmental feedback can reduce employees’ time theft through serial multiple mediating effects of employees’ perceived insider status and work passion. Study 2 employed topic analysis to uncover the topic structure of supervisor developmental feedback text, which includes skill learning, attitude learning, and social learning. Study 2 further affirms the serial multiple mediation model across different supervisor developmental feedback topics; however, the moderation effect of perceptions of leader hypocrisy is not supported in either studies. In conclusion, our studies validate that supervisor developmental feedback plays an effective preventive role in addressing employees’ time theft.

### 5.2. Theoretical Contributions

First, this study investigated the preventive effect of leaders’ developmental feedback on employees’ time theft and analyzed both behavioral data and textual data of supervisor developmental feedback. Existing studies at the leadership level have mainly examined the relationship between leadership styles and time theft, e.g., authoritarian leadership [16] and laissez-faire leadership [19]. However, leadership styles are highly conceptualized and relatively stable, resulting in difficulty in real interventions. For example, both authoritarian leadership and laissez-faire leadership can exacerbate employees’ time theft [49,50], placing leaders in a dilemma that they should avoid being either too strict or too lenient. Additionally, leadership styles are often associated with various deviant behaviors, e.g., authoritarian leadership has a positive correlation with broader employee deviant behavior [49], meaning that it is often difficult to merely employ leadership style to identify the corresponding and effective time theft preventive methods. In light of the considerations mentioned above and the fact that leaders typically influence employees through communications [22], our study introduces the concept of supervisor developmental feedback, which refers to leaders’ behavior in providing helpful and valuable feedback information to employees with the aim of assisting their future learning and growth [23]. As hypothesized, the results suggest that supervisor developmental feedback can reduce employee time theft. According to role theory, supervisor developmental feedback can reduce work role ambiguity as well as motivate employees to increase work role centrality. Ultimately, employees may proactively perform the work role more appropriately and conduct less time theft. These findings enrich the research on preventing employee time theft through leaders’ behaviors. Apart from examining behavioral intensity, an existing research gap lies in supervisor developmental feedback content analysis [23]. Previous studies have provided examples of supervisor developmental feedback [24], whereas there is a vacancy for a systematic topic structure. Hence, Study 2 of this research employed natural language processing techniques to conduct a topic analysis of supervisor developmental feedback. The results indicate that supervisor developmental feedback consists of three main topics, namely, skill learning, attitude learning, and social learning. Furthermore, the hypotheses of the serial multiple mediation model were examined across topics. Although there were no significant differences in outcomes related to different topics in this study, the topic structure laid the foundation for a more in-depth investigation into the mechanisms of subsequent supervisor developmental feedback studies.

Second, this study constructed a serial multiple mediating mechanism to reduce time theft based on role theory. The mechanism involves ‘supervisor developmental feedback–perceived insider status–work passion–time theft’. Current studies on time theft have mainly built on the conservation of resources theory, aiming to identify predictors and control them in order to prevent time theft. However, the pervasive and covert nature of time theft makes it challenging to intervene [2,4]. In addition, multiple antecedents of time theft may lead to a leader’s behavioral dilemma, as mentioned above. Therefore, scholars are attempting to figure out a direct mitigation mechanism for time theft [19]. Based on role theory [21], the present study incorporates perceived insider status and work passion as mediating variables in the relationship between supervisor developmental feedback and time theft, reflecting the ‘recognizing work role–internalizing work role to selfconcept–behavior’ process by which employees internalize organizational expectations. Additionally, prior supervisor developmental feedback research has mainly focused on employees’ positive outcomes, e.g., job performance [24], whereas the effect on mitigating employee deviant behaviors has received limited attention. Our study adds further evidence to the impact of supervisor developmental feedback on deviant behavior.

Finally, this study explore the boundary effect of inconsistency on the behaviors and statements present in supervisor developmental feedback. The majority of previous studies on supervisor developmental feedback have mainly focused on employees’ traits, e.g., emotional intelligence [61], whereas few have considered leaders’ traits. In addition, scholars in leadership studies have focused more on the ambivalent effect of leadership styles in recent years, e.g., the attribution of leadership humility to impression management tactics, where leaders’ attempts to be seen favorably or positively [49], and the attribution of authenticity to self-sacrificing leadership [50]. Thus, our paper examined the moderating effect of perceptions of leader hypocrisy based on the potential inconsistency in the behaviors and statements of supervisor developmental feedback. However, the moderation effect did not receive support in this study. This may be due to three reasons. First, leaders often categorize employees into different ‘circles’ and cultivate a subset of employees as personal ‘insiders’ at work [44]. Even in the case of a hypocritical leader, employees may still perceive themselves as being treated as ‘insiders’ by the leader, leading to a certain level of trust despite the perceived dishonesty. As leaders hold significant influence in organizations [33], employees may believe they are objectively considered as ‘insiders’ by the leader when receiving developmental feedback, leading to organizational recognition and a positive work attitude. Second, although supervisor developmental feedback is delivered by leaders, it may not solely reflect their personal intentions, and may encapsulate organizational beliefs as well. In such instances, even if employees perceive leaders as hypocritical, the very provision of developmental feedback acts as an indicator that the organization values them, leading to an enhanced sense of organizational identity. Within organizations, employees often exert effort to better fulfill their roles rather than to seek individual rewards from leaders. Therefore, even if the inconsistency between leaders’ words and actions may cause the leader–employee relationship to deteriorate, employees might maintain a sense of being organizational insiders in response to supervisor developmental feedback. Lastly, even if leaders do not require employees to achieve specific task goals [23], they may still point out specific paths, that is, skill learning, attitude learning, and social learning. Hypocritical leaders may choose to provide more outlooks and encouragement for employees’ futures without specifying explicit action plans. Regardless of whether leaders provide clear action plans in developmental feedback, employees still perceive that their growth is valued by the organization. From a cost–benefit perspective, the helpful information from supervisor developmental feedback is more beneficial to employees. Thus, even if employees perceive their leaders as hypocritical overall, they remain willing to respond positively to supervisor developmental feedback.

### 5.3. Practical Implications

Nowadays, the evolution of network technology has intensified the ‘cat-and-mouse game’ between organizations and employees in combating time theft, particularly when employees were sent home to work remotely due to the COVID-19 pandemic. While employees have more technical ways to occupy working time, e.g., engaging in online shopping during work hours [4], many companies are attempting to develop or introduce advanced employee behavior monitoring techniques (e.g., the live views of employee PCs described in the case above) to curb employee time theft. However, scholars have affirmed that these measures not only trigger employee dissatisfaction [15] but also incur additional capital and labor costs, even triggering legal disputes [5]. Our study proposes that a slight change in a leader’s daily behavior, specifically through developmental feedback, can make a difference in employees’ time theft. In detail, employees will proactively reduce their time theft when leaders more frequently provide feedback information with the aim of improving employees’ learning and future growth, such as learning specific skills, improving work attitudes, and studying role models within the organization. While previous studies have suggested that leaders should neither be too strict nor too lenient concerning time theft [16,19], our study points out an effective way to prevent employee time theft without placing leaders in a behavioral dilemma. From a broad perspective, our study reveals that, beyond punitive measures and controlling triggers, support and guidance from leaders can motivate employees to take the initiative in reducing time theft.

### 5.4. Limitations and Directions for Future Research

This study employed employee self-reports to collect cross-sectional data for variables, as reliably gauging how others perceive employees’ time theft is challenging. Although the empirical results indicate that method bias in the data is not a significant problem, future research could use varied reporting sources for the collection of longitudinal data. Additionally, prior research has revealed that employees may overlook instances of time theft that occurred long ago [53]. Moreover, social desirability cannot be completely eradicated, even though its influence on measuring employee time theft is negligible [3]. Thus, with ethical approval, future research could explore objective data collection methods for time theft, such as work system time records. Furthermore, daily experimental designs could be considered to study how supervisor developmental feedback affects employee time theft from the standpoint of emotional fluctuations. Scholars have found that employees with negative emotions tend to engage in more internet-based work procrastination [16]. Thus, future research can establish an emotion-centered model and explore the coordinated relationship among supervisor developmental feedback, emotion, and time theft on a daily basis.

Concerning feedback text topic analysis, this study used the SBERT-LDA model, as the feedback consisted of short sentences rather than long texts. Previous studies have shown that this model outperforms other topic models such as BTM, LDA, and Word2Vec in terms of topic consistency [58]. For the main purpose of Study 2 was to identify feedback topics, this study did not delve into evaluating the results from different topic models. In future research, scholars could use topic consistency and other indicators to assess and compare topic models, and even develop language models tailored for workplace communication.

Study 2 explored the question “What is the content of supervisor developmental feedback?” However, the question “How is the developmental feedback communicated?” is currently changing due to the rise of remote work. Leaders now use various feedback channels, including but not limited to email, phone, video conference, and other tools, to deliver feedback to employees [62]. Traditionally, employees have preferred to receive developmental feedback from leaders face-to-face instead of through electronic channels, as it often pertains to personal and confidential information [63]. However, the shift towards remote work spatially separates leaders from employees, prompting a need to understand the effectiveness of different supervisor developmental feedback channels. Existing research on feedback channels has mainly focused on performance feedback or general communication, while a controversy lies in the efficacy of feedback channels. Scholars have argued that face-to-face feedback can lead to higher perceived quality of communication and employee work satisfaction than phone or email, as face-to-face feedback tends to be more personalized and able to convey more social information through the leader’s tone of voice, body language, etc. [64]. On the other hand, scholars have used laboratory experiments to compare the effects of computer-mediated, text message, and face-to-face performance feedback on lateral task performance and found no significant difference in performance improvements across these channels [65]. Considering that our study has identified three categories of supervisor developmental feedback using textual analysis, future research could further investigate whether supervisor developmental feedback across different topics holds when distinguishing channel adaptability. For instance, due to the relatively distinctive and consistent nature of skills, skill learning-oriented supervisor developmental feedback may be less sensitive to the choice of feedback channel than the other two forms, which require more social information. 

## Figures and Tables

**Figure 1 behavsci-14-00269-f001:**
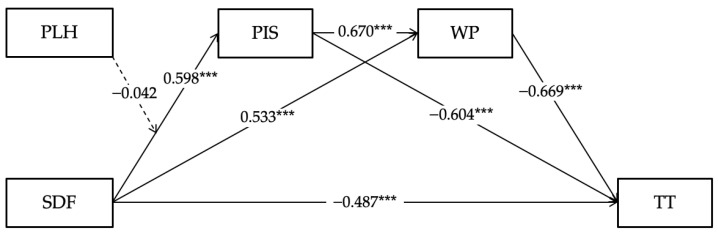
Notes: The model includes the results of the mediation analysis and the moderating effect of the interaction term between supervisor developmental feedback and perceptions of leader hypocrisy in the first stage; standardized coefficients (β) are displayed. Considering simplicity of the graphic, the path coefficients for the control variables are not presented. SDF = supervisor developmental feedback; TT = time theft; PIS = perceived insider status; WP = work passion; PLH = perceptions of leader hypocrisy; *** p<0.001.

**Figure 2 behavsci-14-00269-f002:**
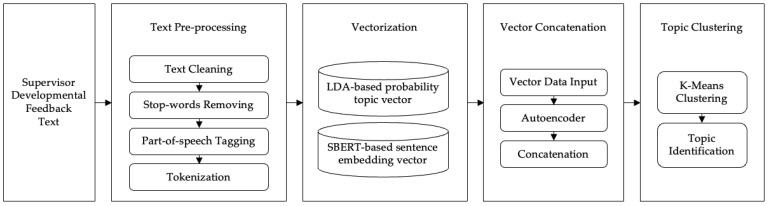
An integrated architecture of SBERT and LDA model for supervisor developmental feedback text topic recognition.

**Figure 3 behavsci-14-00269-f003:**
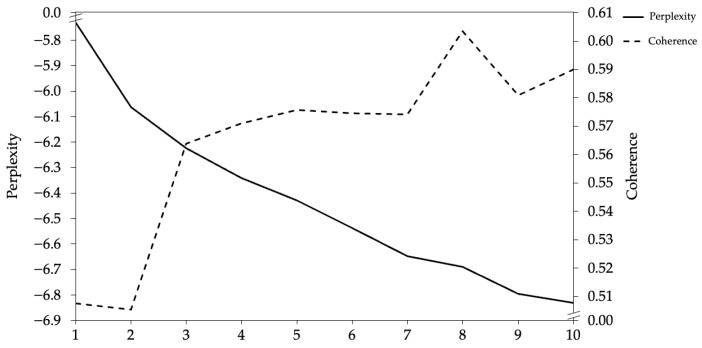
Topic perplexity and coherence.

**Figure 4 behavsci-14-00269-f004:**
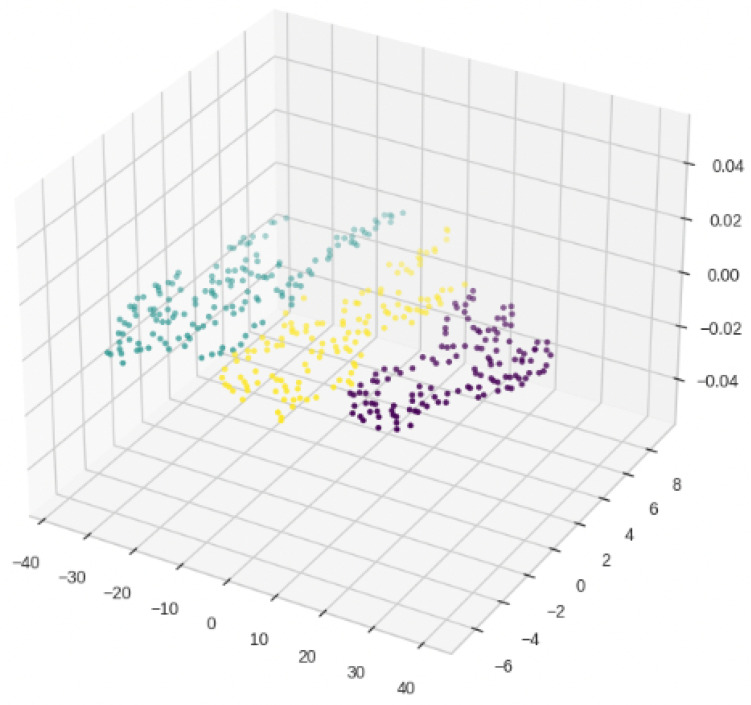
Topic clusters distribution.

**Table 1 behavsci-14-00269-t001:** Results of confirmatory factor analysis.

Model	$\chi^{2}$	df	$\chi^{2}$/df	CFI	TLI	SRMR	RMSEA
5-factor model (S, I, W, H, T)	533.084	179	2.978	0.897	0.879	0.055	0.070
4-factor model (S, I+W, H, T)	585.223	183	3.198	0.883	0.865	0.054	0.074
3-factor model (S+I+W, H, T)	658.516	186	3.540	0.862	0.845	0.059	0.079
3-factor model (S, I+W+H, T)	777.726	186	4.181	0.828	0.805	0.063	0.089
2-factor model (S+I+W+H, T)	807.519	188	4.295	0.819	0.798	0.065	0.091
2-factor model (S, I+W+H+T)	837.855	188	4.457	0.811	0.788	0.066	0.093
1-factor model (S+I+W+H+T)	856.941	189	4.534	0.803	0.781	0.066	0.094

Notes: S refers to supervisor developmental feedback; I refers to perceived insider status; W refers to work passion; T refers to time theft; H refers to perceptions of leader hypocrisy.

**Table 2 behavsci-14-00269-t002:** Descriptive statistics and correlations.

Variable	M	SD	1	2	3	4	5	6	7	8	9	10	11
1. Gender	0.64	0.48											
2. Age	2.61	0.80	−0.02										
3. Education	3.04	0.70	0.07	−0.06									
4. Job type	3.45	1.10	−0.12 *	−0.02	0.02								
5. Industry	6.25	3.75	0.09	−0.07	−0.05	−0.09							
6. Tenure	3.21	1.25	−0.04	0.75 **	−0.02	−0.05	−0.13 *						
7. Job type	1.99	1.09	0.11 *	0.29 **	0.25 **	−0.02	−0.08	0.43 **					
8. SDF	5.34	0.76	−0.05	0.15 **	−0.02	0.03	−0.11 *	0.24 **	0.15 **				
9. TT	2.39	0.98	0.03	−0.19 **	0.06	−0.06	0.13 **	−0.18 **	−0.11 *	−0.51 **			
10. PIS	5.38	0.76	−0.10	0.21 **	−0.08	0.01	−0.10 *	0.29 **	0.16 **	0.64 **	−0.61 **		
11. WP	5.42	0.70	−0.05	0.25 **	−0.08	0.04	−0.18 **	0.28 **	0.12 *	0.58 **	−0.67 **	0.70 **	
12. PLH	3.03	0.97	0.00	−0.14 **	0.02	0.01	0.20 **	−0.16 **	−0.12 **	−0.59 **	0.53 **	−0.49 **	−0.50 **

Notes: N = 402; SDF = supervisor developmental feedback; TT = time theft; PIS = perceived insider status; WP = work passion; PLH = perceptions of leader hypocrisy; * p<0.05, ** p<0.01.

**Table 3 behavsci-14-00269-t003:** Results of hierarchical regression analysis.

	PIS		WP			TT					
	**1**	**2**	**3**	**4**	**5**	**6**	**7**	**8**	**9**	**10**	**11**
Gender	−0.061	−0.068	−0.0002	0.035	0.031	−0.005	−0.037	−0.034	0.002	−0.006	−0.020
Age	0.010	−0.010	0.107	0.093	0.101	−0.159 *	−0.146 *	−0.154	−0.081	−0.099	−0.109 *
Education	−0.068	−0.073	−0.063	−0.022	−0.026	0.055	0.019	0.022	0.018	0.020	0.011
Job type	−0.010	−0.004	0.020	0.031	0.026	−0.038	−0.048	−0.042	−0.028	0.026	−0.031
Industry	−0.012	0.011	−0.013 *	−0.105 *	−0.097 **	0.073	0.075	0.067	0.014	0.014	0.023
Tenure	0.120	0.130 *	0.070	0.020	0.005	0.074	0.118	0.132	0.090	0.114	0.134 *
Job level	0.036	0.043	−0.014	−0.028	−0.033	−0.029	−0.016	−0.012	−0.048	−0.037	−0.026
SDF	0.598 ***	0.511 ***	0.533 ***		0.210 ***	−0.487 ***		−0.199 ***		−0.186 ***	−0.105 *
PIS				0.670 ***	0.540 ***		−0.604 ***	−0.481 ***			−0.239 ***
WP									−0.669 ***	−0.564 ***	−0.449 ***
PLH		−0.185 ***									
SDF×PLH		−0.042									
$R^{2}$	0.438	0.460	0.376	0.514	0.540	0.283	0.389	0.412	0.459	0.481	0.505
*F*	38.264 ***	33.358 ***	29.638 ***	52.000 ***	51.123 ***	19.350 ***	31.311 ***	30.566 ***	41.639 ***	40.413 ***	39.871 ***

Notes: Standardized coefficients (β) are displayed; SDF = supervisor developmental feedback; TT = time theft; PIS = perceived insider status; WP = work passion; PLH = perceptions of leader hypocrisy; * p<0.05, ** p<0.01, *** p<0.001.

**Table 4 behavsci-14-00269-t004:** Mediating effects with 95% confidence intervals using bootstrap method.

Path	Direct Effect	95% CI	Indirect Effect	95% CI
SDF→PIS→TT	−0.199	[−0.299, −0.099]	−0.288	[−0.376, −0.210]
SDF→WP→TT	−0.185	[−0.274, −0.097]	−0.301	[−0.375, −0.236]
SDF→PIS→WP→TT	−0.105	[−0.199, −0.011]	−0.382	[−0.477, −0.302]

Notes: Standard effects are displayed; SDF = supervisor developmental feedback; TT = time theft; PIS = perceived insider status; WP = work passion; CI = confidence interval.

**Table 5 behavsci-14-00269-t005:** Topics of supervisor developmental feedback.

Topic	N	High-Frequency Topic Words
Skill learning oriented SDF	126	Competence; Level; Skill; Knowledge
Attitude learning oriented SDF	148	Mind; Quality; Experience; Effort
Social learning oriented SDF	128	Colleague; Occupation; Advantage; Information

Notes: SDF = supervisor developmental feedback; high-frequency topic words are not a complete set.

**Table 6 behavsci-14-00269-t006:** Results of hypothesis testing across supervisor developmental feedback topics.

Independent Variable	Serial Multiple Mediating Effect	Moderation Effect	Estimation Diagram
Skill learning oriented SDF	Total indirect effect is sig. (β = −0.379, 95% CI = [−0.536, −0.253])	Non-sig.	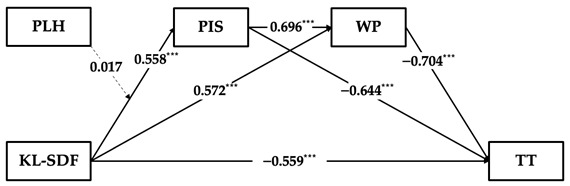
Attitude learning oriented SDF	Total indirect effect is sig. (β = −0.405, 95% CI = [−0.571, −0.266])	Non-sig.	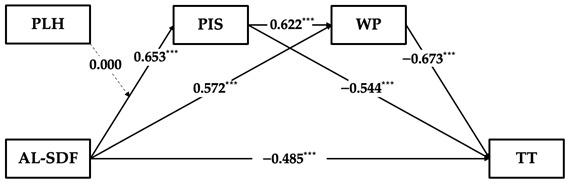
Social learning oriented SDF	Total indirect effect is sig. (β = −0.352, 95% CI = [−0.552, −0.215])	Non-sig.	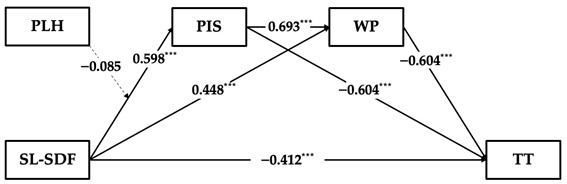

Notes: SDF = supervisor developmental feedback; KL-SDF = skill learning-oriented supervisor developmental feedback; AL-SDF = attitude learning-oriented supervisor developmental feedback; SL-SDF = social learning-oriented supervisor developmental feedback; TT = time theft; PIS = perceived insider status; WP = work passion; PLH = perceptions of leader hypocrisy; *** p<0.001.

## Data Availability

Research data of this study can be obtained from the corresponding author upon reasonable request.

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
