# Peer review of "Reducing Employees’ Time Theft through Leader’s Developmental Feedback: The Serial Multiple Mediating Effects of Perceived Insider Status and Work Passion"

_behavsci, 2024, doi:10.3390/bs14040269_

Round 1

Reviewer 1 Report

Comments and Suggestions for Authors

Goal: this article aims to assess leader development feedback on employees’ behaviors regarding time theft using new data.

Strengths: the article explores an interesting subject.

Weaknesses: some that are listed below.

Therefore, I request a minor revision of the paper that fixes the following issues.

 Major remarks:

 - Introduction: the authors should give more examples of time theft. They need, for instance, to cover cyberloafing and cite some papers such as

Ma L., Zhang X, Yu P. (2023) Enterprise social media usage and social cyberloafing: an empirical investigation using the JD-R model. Internet Res.

Koay K.Y. (2018) Workplace ostracism and cyberloafing: a moderated –mediation model. Internet Res. 28(4):1122-41.

Corgnet B., Hernán-González R., & McCarter M. (2015). The role of the decision-making regime on cooperation in a workgroup social dilemma: An examination of cyberloafing. Games, 6(4), 588–603.

- The authors claim that their sample is well representative. To prove this, they should provide a table comparing at least gender, age and industry with external official information using a t-test. And if there are differences, they should reweight their data using, for instance, the weighting procedure available on Stata with a package ‘ipfraking’ developed by Kolenikov (2014).

Kolenikov, S. (2014). Calibrating survey data using iterative proportional fitting (raking). The Stata Journal, 14(1), 22–59.

- The authors should add a table in the appendix listing all the dimensions of items used in their paper.

- for the K-means, the authors should provide information on the way they choose the number of clusters like a dendrogram (see Appendix Figure 1) and the ‘elbow’ method that identify a slope reduction (or a stable or an increase) at a point on the graphic of the sum of the squared distances.

- The end of the conclusion is somewhat abrupt. The authors should end by paving the way for future research, for example, on assessing the role of the means of communication between managers and employees in the way managers give feedback (face-to-face, e-mail, videoconferencing, individual or team discussion).

Minor remarks:

 - The authors should tell in the abstract that the same sample is used for Study 1 and Study 2.

Author Response

Dear Reviewer,

We wish to express our gratitude for the thorough review of our manuscript and the constructive, developmental feedback you have provided. Please find attached our response letter detailing the actions taken and our responses to your comments.

With best regards,

The Authors

Reviewer 2 Report

Comments and Suggestions for Authors

The article addresses an important topic.
At the same time - in the context of the formulated aim - in my opinion the theoretical introduction is too general. It is not clear what the factors influencing theft are. They have not been shown. Nor have the reasons for focusing attention on selected (analysed) ones been explained.
Either the content should be supplemented with the above or the study aim should be reformulated accordingly.

Furthermore:
1. Hypothesis 4 and 3 have the same wording
2. I have doubts about the conduct of the study. This is because there is a lack of information regarding:
how the study participants were reached (were they invited? where was the questionnaire placed?)
how were they selected? (were there specific criteria?)
for what reason was the sample considered balanced, if, for example, by age we have a predominance of people between 20 and 40? are they the ones affected by the phenomenon under investigation? How do you know this?

Author Response

(The authors gave the same response as above.)

Reviewer 3 Report

Comments and Suggestions for Authors

The paper starts by showing the relevance of the topic through referring to a cost of $759 billion. However, the reference they give is paper which itself is just citing a newspaper source. Therefore, we do not have any scientifically relevant indication. This is not good practice. You should justify relevance otherwise. Especially, to label activities not directly related to the task at hand as theft or banditry, as the other paper does, is very harsh and potentially not doing justice to the potentially indirect positive aspects of such activities (learning, communication etc.). Lacking a clear justification for this, your whole endeavor, looking for ways to reduce it, is undermined. One way to overcome this, would be to replace the negative view on time theft through a more neutral stance.

You cite Lorinkowa et al (2014) who did already use the concept of time theft, but have a much more nuanced look on it, for example, acknowledging the fact, that time theft may be a means for reducing frustration, for example, with leaders. Thus, while they see time theft as a possible reaction to leader behaviors (via cynism) you are using leader behaviors as ways to reduce or mediate time theft. Thus, your hypotheses do substantially deviate from their view. You should explicitely acknowledge this and justfify it. Otherwise your theory is substantially lacking backing. Despite this essential lack, development of hypotheses seems sound, only I do not understand hypothesis 6, which seems to be just added at convenience. Please provide further explanation for this.

Method

I was rather confused in regard to the mixing of hierarchical regression analysis and the bootstrapping. Why are you doing this. Please give more justification for this. Otherwise it is good practice to stay with the most concise method of analysis, which is in this case hierarchichal regression. Further, while I appreciate the use of text-based analysis, the additional value of Study 2 is also rather unclear. Please justify this more.

Author Response

(The authors gave the same response as above.)

Reviewer 4 Report

Comments and Suggestions for Authors

The interesting paper addresses an issue that has been discussed for long in the context of communist countries (without a free market economy), especially in the Soviet Union (e.g. Paul R. Gregory: Productivity, Slack, and Time Theft in the Soviet Union, Houston University 1986).  

The authors decide for a framework not directly embedded in modern institutional economics, e.g. based on principal-agency analysis which potentially could enrich the analysis. The same could be said for including "efficiency wage" theories which precisely address the issue of motivating employee efforts and to align the incentives of all sides ("win-win-incentives") within the company between owners and employees or within a hierarchy where you often have also heads of departments/sections etc. (The article refers to the relationship between employee(s) and "leader" and this context is used as the background for discussion.) Time theft could be minimised by avoiding incentives for such "theft". 

Whether the value-lade term should be used in that way could be debated since in much of the "personnel literature" it is used only for e.g. property or software theft. This is, perhaps, because one has to define properly what is meant by such theft (lines 72-74). More should be discussed here sind the debate of this is very limited and potentially misleading. Especially when one speaks about "unsanctioned non-work related activities". E.g.how can it  be "theft" if it is in line with what is typically allowed or at least accepted ("unsanctioned")? Modern workplaces even in services can differ a lot regarding the (technical) monitoring opportunities, and more differentiation and discussion on this would be worthwhile.

The "story" of the article becomes quite clear rather at the end only (pages 16-17) when the article more practically explains the results and implications by spelling out clearly that largely independent of other factors (especially hypocrisy of "leaders") a positive response to supervisor's development feedback does usually exist (line 646). Other causes that may explain the "exercise of self-restraint in time theft to fulfil their work responsibilities" (line 638) remain, however, quite vague (even though they may be very important). 

Finally, certain terms English could be explained clearer by the authors. e.g. "impression management strategy" since this technical term may be defined differently by different academics. Moreover, the often used verb "moderate" should be replaced by a perhaps a clearer causal term, e.g. "to affect" and by telling the direction how the expected relationship is. 

Comments on the Quality of English Language

See last paragraph above. English can, however, be quite well understood by a non-native academic. However, some terms could be used in a more precise way and definitions of technical terms should be given (rather than only quoting a source where a definition is included). Sometimes. there may be also competing definitions and more space could be used to discuss potentially controversial uses of such technical terms.

Author Response

(The authors gave the same response as above.)

Reviewer 5 Report

Comments and Suggestions for Authors

Modern work processes are changing under the influence of both macro factors, such as: globalization; pandemic; problems of ecological changes, such as the effects of warming or logistical problems caused by local wars or riots; technological progress. This results in the use of various forms of flexibility at the micro level, i.e. at workplaces.

   In particular, the implementation of the lockdown during the COVID-19 pandemic resulted in a forced increase in remote work at home. As a result, new problems emerged, such as managers' inability to directly control employees' work, procrastination, and time theft. Taking up the problem of time theft, the authors addressed a theoretically and practically important research gap.

 By defining time theft as: "employees' tendency to engage in unsanctioned activities unrelated to work during working time", the concept of role theory was adopted as the basis for research (professional vs. non-professional roles). The theoretical gap causes a quite wide range of considerations to be adopted in two studies. Various aspects of the role of feedback on the supervisor's development were analyzed, and 6 hypotheses were proposed regarding the influence of the moderating effects of outsider's and insider's actions, passion for work and leader's hypocrisy between developmental feedback and time theft.

The quantitative study used descriptive statistics and correlation analysis (means, standard deviations and intercorrelations between measured variables); hierarchical regression analysis using SPSS 27.0; Hayes' Model 6 of the SPSS PROCESS; SPSS PROCESS Model 1. The qualitative study (feedback texts) used the SBERT-LDA model, i.e. integrating latent Dirichlet allocation (LDA) with bidirectional representations of the sentence encoder from transformer addresses (SBERT). The selection of survey questionnaires, research methods and techniques, and the broadcast sample are methodically correct and statistically verified (including Cronbach's alpha).

 The research results demonstrate the rejection of two hypotheses, namely: (5) Perceptions of leader hypocrisy negatively moderates the relationship between supervisor developmental feedback and perceived insider status, and (6) Perceptions of leader hypocrisy negatively moderates the serial multiple mediating effects of supervisor developmental feedback on time theft via perceived insider status and work passion. Since the presented research is only just opening the possibility of verifying the factors influencing time theft, this does not mean at this stage questioning the importance of the leader's hypocrisy. Attention should be paid to the high cognitive value of the research, the novelty of the topic, as well as a sensible and interesting attempt to fill the research gap.

Author Response

(The authors gave the same response as above.)
